# Bilateral Tuberculum Sextum of Maxillary Permanent First Molar

**DOI:** 10.3390/diagnostics15020134

**Published:** 2025-01-08

**Authors:** Mrunali Prashant Deshkar, Yash Naik, Ramakrishna Yeluri, Nilima Thosar, Monika Khubchandani, Meenal Pande

**Affiliations:** Department of Pediatric and Preventive Dentistry, Datta Meghe Institute of Higher Education and Research, Wardha 442001, Maharashtra, India; yashyogiminanaik4@gmail.com (Y.N.); ramakrishna.pedo@dmiher.edu.in (R.Y.); nilima.pedo@dmiher.edu.in (N.T.); monika.pedo@dmiher.edu.in (M.K.); meenal.pedo@dmiher.edu.in (M.P.)

**Keywords:** tubercle, cusp, permanent molar, occlusion, tuberculum sextum

## Abstract

The human tooth’s morphology, which includes variations in cusp numbers and patterns, is of tremendous interest to anthropologists, morphologists, and dentists. Cusp 6 is an additional cusp that is very seldom encountered in primary or permanent mandibular molars, especially first molars. A supernumerary cusp located lingual to the distobuccal cusp at the crown’s distal border is cusp 6. According to the literature, cusp 6 is also known by other anthropologic designations, such as “Tuberculum Sextum” or “Entoconulid”. This case offers a unique instance of a bilateral tuberculum sextum in the maxillary permanent first molars, characterized by an additional cusp on the lingual surface. The patient, an adolescent, exhibited no associated symptoms. Early detection of such dental anomalies is essential for treatment planning, particularly in maintaining occlusal balance and preventing future complications. Regular monitoring is necessary to manage a bilateral tuberculum sextum in order to avoid enamel fractures and occlusal interference. To preserve functional occlusion and avoid problems, prophylactic sealants or selective grinding may be taken into consideration. This report highlights the importance of recognizing a tuberculum sextum for proper clinical management.

**Figure 1 diagnostics-15-00134-f001:**
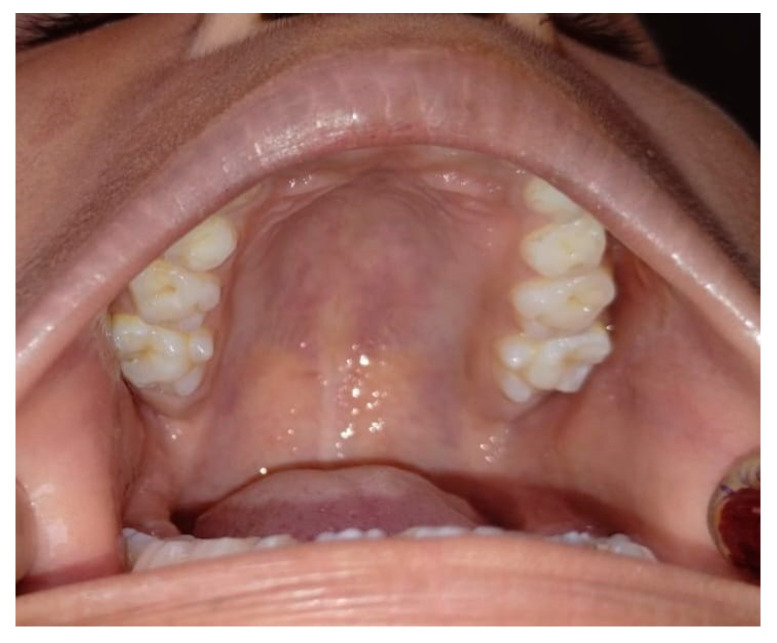
Bilateral tuberculum sextum of maxillary permanent first molar. In a camp at the Adivasi Ashram School in the Wardha District, a 12-year-old male patient was examined. The patient presented with no significant medical history, and their primary complaint was unrelated to the presence of the paramolar tubercle as it was an incidental finding during a routine examination at a community dental camp. The clinical examination revealed no signs of caries, pulpal involvement, or periodontal concerns. It was discovered that the patient had a tuberculum pulpale located on the mesiobuccal cusp of their maxillary right second permanent tooth. The tubercle had a well-formed cusp and a pointed tip. There were no soft-tissue anomalies found. Bilateral extra cusps were visible on the mesiobuccal cusps of 16 and 26 (Figure 1). Every extra cusp was out of occlusion and had a groove separating it from the corresponding tooth. The most frequently reported extra or accessory cusps are the cusp of Carabelli, the Talon cusp, and Leong’s tubercle, each with distinct characteristics and locations. The cusp of Carabelli, observed in 52–68% of cases, is an additional fifth cusp or tubercle located on the mesiopalatal cusp of maxillary permanent and deciduous molars. The Talon cusp, with a prevalence ranging from 1 to 7.7%, is a morphological developmental anomaly that appears as a cusp-like projection originating from the cingulum area of maxillary or mandibular anterior teeth, resembling the talon of an eagle. Leong’s tubercle, found in about 8% of cases, refers to a unique morphological feature on premolars characterized by an extra “horn” or projection on the biting surface. These accessory cusps contribute to the anatomical variations of teeth, often requiring clinical attention due to their implications for dental function and aesthetics [1,2]. The supplementary cusp creation on the buccal surfaces of the top and bottom everlasting molars in humans was first described by Bolk in 1916, and this cusp was assigned the name “paramolar tubercle” [3,4]. “Paramolar tubercle” is the term used to describe any stylar aberrant cusp, supernumerary inclusion, or elevation that appears on the buccal surfaces of both the upper and lower premolars and molars. Orally and anthropologically, these physical irregularities are quite important. These additional cusps can lead to dental issues like cavities in the crevices or grooves between the extra cusp and the tooth, changes in jaw alignment from premature tooth contact, sensitivity or nerve damage from breakage or wear on the added cusp, and other complications. A paramolar tubercle can disrupt normal occlusion, leading to issues like occlusal trauma and tooth sensitivity. Its grooves are plaque-retentive, increasing caries risk, and fractures may expose the pulp, causing pain or infection. It complicates periodontal health, prosthodontic designs, and orthodontic planning. Early diagnosis and management, including preventive measures like sealants, are essential. In severe cases, surgical intervention may be necessary to address functional or aesthetic concerns. Therefore, appreciating anatomical differences is quite important [4,5]. Paramolar tubercles require careful clinical and radiographic evaluation to assess their size, location, and impact on occlusion and the surrounding structures. Clinically, they may cause occlusal interference, increase caries risk, or lead to pulpal or periodontal issues. Radiographs help to confirm their presence, assess the proximity to the pulp, and evaluate the anatomical implications. Treatment planning should include preventive measures, restorative modifications, and occlusal adjustments, with surgical removal considered for severe cases. Early diagnosis ensures functional, aesthetic, and long-term management [6]. This case report’s objective is to raise awareness regarding the unintentional discovery of a rare dental characteristic in permanent dentition called a paramolar tubercle, often known as Bolk’s cusp. In conclusion, the paramolar tubercle can affect occlusion, increase caries risk, and complicate restorative treatments. The lack of radiographs in this case due to the community dental camp setting highlights the importance of comprehensive radiographic evaluation in clinical practice. Early identification enables better management and prevention of complications. Future studies should focus on the prevalence, long-term effects, and treatment protocols for paramolar tubercles. Standardized diagnostic and management guidelines are needed to improve care for affected patients.

## Data Availability

No new data were created or analyzed in this study.

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
