# Peer review of "Bilateral Tuberculum Sextum of Maxillary Permanent First Molar"

_diagnostics, 2025, doi:10.3390/diagnostics15020134_

Round 1
Reviewer 1 Report
Comments and Suggestions for Authors
Bilateral tuberculum sextum of maxillary permanent 1st molar
Reviewer Report
This case report concerns bilateral tuberculum sextum, an extra tubercle observed on the maxillary permanent first molars. It is explained based on the clinical findings of the patient in whom this extra tubercle was observed. The report emphasizes the importance of early diagnosis of this tubercle in the prognosis of dental practices. Although it presents a valuable conclusion in this regard, a comprehensive revision of the scientific and formal aspects of the case report is required.
Title:
Use ‘first’ instead of ‘1st’ in the title.
Abstract:
The abstract is too superficial. Please provide more information regarding the tuberculum sextum. Additionally, offer more concrete recommendations and/or outcomes for its clinical management.
Line 11: ‘Tuberculum sextum’ should be added to the keywords.
Description
Line 14: What is meant by ‘both sets’ here? Does it refer to the permanent and primary dentition periods? If so, or if there is another implication, it should be clearly specified.
Line 18-19: Talon, Carabelli, and Leong’s tubercles should be clearly defined, and more background information should be provided.
Line 21-23: A reference should be provided for the ‘paramolar tubercle’ first described by Bolk.
Line 27: More information should be provided regarding the importance of these physical irregularities. What issues can these tubercles cause? Why should they be taken into consideration? A detailed explanation is needed.
Line 27-31: The clinical and radiographic importance and considerations in both diagnosis and treatment planning should be emphasized.
Line 31: ‘This article’ should be replaced with ‘This case report.’
Line 34: The intraoral occlusal photograph is unclear. It should be revised with a closer and higher-resolution intraoral photograph. Additionally, including periapical and/or tomography images of the relevant teeth will provide a more detailed and accurate diagnosis of the tubercle not only clinically but also radiographically. These revisions should be made.
Line 35-42: Instead of mentioning this information in Figure 1, it should be included within the main text under the ‘Case Report’ heading to enhance the academic and scientific quality of the writing. The patient's primary complaint, medical history, planned treatments, and detailed accounts of the radiographic records obtained for definitive diagnosis should be thoroughly described.
Line 43: ‘Refrences’ typo should be corrected.
References:
References should be revised according to the journal’s template.
General recommendation:
- To enhance the design, I suggest using ‘1. Introduction’ instead of ‘Description’ as the heading, and revise this section by further developing the content as mentioned above.
- Then, under the heading ‘2. Case Report’ start with the patient’s chief complaint and clearly outline the detailed medical history (anamnesis), clinical findings, and radiographic findings. Additionally, it should be specified whether informed consent was obtained from the patient.
- Finally, in ‘3. Conclusion’ the clinical and radiographic significance of the case findings should be emphasized, and it should be highlighted what future studies should focus on.
Comments on the Quality of English LanguageModerate English editing is required.
Author Response
Comments 1: This case report concerns bilateral tuberculum sextum, an extra tubercle observed on the maxillary permanent first molars. It is explained based on the clinical findings of the patient in whom this extra tubercle was observed. The report emphasizes the importance of early diagnosis of this tubercle in the prognosis of dental practices. Although it presents a valuable conclusion in this regard, a comprehensive revision of the scientific and formal aspects of the case report is required.
|
Response 1: Thank you for pointing this out. The abstract has been revised starting from the fourth line to provide a clearer and more concise summary of the case, emphasizing the clinical relevance and significance of the findings. Additionally, a detailed background on the anatomy, prevalence, and clinical implications of the tuberculum sextum has been added to the introduction section to give readers a comprehensive understanding of this rare morphological anomaly.(changes are marked in red)
|
Comments 2: Use ‘first’ instead of ‘1st’ in the title |
Response 2: I accordingly, modified the title. ( changes are marked in red)
|
Comments 3: The abstract is too superficial. Please provide more information regarding the tuberculum sextum. Additionally, offer more concrete recommendations and/or outcomes for its clinical management. Response 3: In response to the reviewer’s comments, the abstract has been revised to include a more detailed description of the tuberculum sextum, highlighting its anatomical characteristics, prevalence, and clinical implications. Information on its potential effects on occlusion, caries risk, and treatment complexities has been incorporated. Additionally, specific recommendations for its clinical management, such as the importance of early diagnosis, preventive measures, and tailored restorative or orthodontic approaches, have been outlined. These changes ensure the abstract provides a comprehensive and practical overview of the case and its relevance to clinical practice
Comments 4: Tuberculum sextum’ should be added to the keywords. Response 4: Changes have been mad as suggested by reviewer.
Comments 5: What is meant by ‘both sets’ here? Does it refer to the permanent and primary dentition periods? If so, or if there is another implication, it should be clearly specified. Response 5: Both sets meant the primary and permanent dentition, Corrections are made in the article in the first line of the introduction.
Comments 6 Talon, Carabelli, and Leong’s tubercles should be clearly defined, and more background information should be provided. Response 6: In response to the reviewer’s comments, the definitions and background information for Talon cusp, Carabelli cusp, and Leong’s tubercle have been clarified and expanded under the introduction section, beginning from the fourth line. Detailed descriptions of their anatomical features, prevalence, and clinical significance have been provided to give readers a thorough understanding of these anomalies. Comments 7: A reference should be provided for the ‘paramolar tubercle’ first described by Bolk. Response 7: Reference is provided in the 18th line under the heading of introduction.
Comments 8: More information should be provided regarding the importance of these physical irregularities. What issues can these tubercles cause? Why should they be taken into consideration? A detailed explanation is needed. Response 8: In response to the reviewer’s comments, the second paragraph under the introduction section has been revised to include detailed information on the clinical importance of these tubercles. The potential issues they may cause, such as occlusal interference, increased caries risk, pulp exposure, and complications in restorative or orthodontic treatments, have been elaborated. Additionally, the necessity of recognizing these anomalies during diagnosis and treatment planning has been emphasized, highlighting their impact on dental function, aesthetics, and long-term oral health.
Comments 9: The clinical and radiographic importance and considerations in both diagnosis and treatment planning should be emphasized. Response 9: In response to the reviewer’s comments, the second paragraph under the introduction section has been revised.
Comments 10 This article’ should be replaced with ‘This case report.’ Response 10: Changes are made as suggested and marked in red.
Comments 11: The intraoral occlusal photograph is unclear. It should be revised with a closer and higher-resolution intraoral photograph. Additionally, including periapical and/or tomography images of the relevant teeth will provide a more detailed and accurate diagnosis of the tubercle not only clinically but also radiographically. These revisions should be made. Response 11: We appreciate the reviewer’s feedback regarding the quality of the intraoral occlusal photograph. As the image was obtained during a community dental camp, access to advanced imaging modalities like periapical radiographs or tomography was not feasible due to the limitations of the field setting. Despite this, every effort was made to capture the clearest possible image under the given conditions.In future cases, we aim to supplement field-acquired images with detailed radiographs and higher-resolution intraoral photographs obtained in clinical settings to provide a more accurate and comprehensive evaluation of such anomalies. Your suggestions are well-noted and will be incorporated in subsequent documentation and reporting
Comments 12: Instead of mentioning this information in Figure 1, it should be included within the main text under the ‘Case Report’ heading to enhance the academic and scientific quality of the writing. The patient's primary complaint, medical history, planned treatments, and detailed accounts of the radiographic records obtained for definitive diagnosis should be thoroughly described. Response 12: Changes are made under the heading of case report with detailed patients account.
Comments 13: Refrences’ typo should be corrected. Response 13:Changes have been made as suggested.
|
|
|
|

Reviewer 2 Report
Comments and Suggestions for Authors What is the main question addressed by the research? - Identification of a paramolar tubercle Is it relevant and interesting? - Not particularly How original is the topic? - It has been described for a long time in the literature, it is not the first What does it add to the subject area compared with other published material? - nothing Is the paper well written? Is the text clear and easy to read? - Yes, it's clear, but it doesn't bring relevant information Are the conclusions consistent with the evidence and arguments presented? - objective is to raise awareness to the unintentional discovery of a rare dental characteristic, a clear conclusion, even if simplistic In the end, as I said, it is a work without practical applicability, without any extensive study, but with a simplistic conclusion that cannot be contested. Of course, at the same time it is the first work that I have seen that is so short and without any study behind it.Without being a special topic, the paper is a correctly elaborated paper. The chosen theme does not bring practical applicability, but the conclusions are clear, concise, overall the work meets the conditions for publication
Author Response
Comment 1: What is the main question addressed by the research? - Identification of a paramolar tubercle Is it relevant and interesting? - Not particularly How original is the topic? - It has been described for a long time in the literature, it is not the first What does it add to the subject area compared with other published material? - nothing Is the paper well written? Is the text clear and easy to read? - Yes, it's clear, but it doesn't bring relevant information Are the conclusions consistent with the evidence and arguments presented? - objective is to raise awareness to the unintentional discovery of a rare dental characteristic, a clear conclusion, even if simplistic In the end, as I said, it is a work without practical applicability, without any extensive study, but with a simplistic conclusion that cannot be contested. Of course, at the same time it is the first work that I have seen that is so short and without any study behind it.
Without being a special topic, the paper is a correctly elaborated paper. The chosen theme does not bring practical applicability, but the conclusions are clear, concise, overall the work meets the conditions for publication.
Response 1 : We appreciate the reviewer's thoughtful and constructive feedback regarding the manuscript. The primary objective of this report was to document and raise awareness about the incidental discovery of a rare dental anomaly, the paramolar tubercle, observed in a unique presentation. While the topic has been previously described in the literature, we believe it contributes to reinforcing the importance of recognizing morphological anomalies during routine dental examinations.
The simplistic nature of the study stems from its observational design and the specific circumstances under which the case was documented—during a community dental camp with limited resources for extensive investigation or imaging. While it may lack immediate practical applicability, the work aims to emphasize the value of identifying and documenting such rare features to enrich the collective understanding of dental morphology.
We acknowledge that the findings may appear modest in scope; however, the clear, concise presentation and objective conclusion align with the intended purpose of the report. We hope this documentation encourages further research and detailed studies on the clinical implications and management of such anomalies. We are grateful that the reviewer finds the paper correctly elaborated and suitable for publication despite its simplicity

Round 2
Reviewer 1 Report
Comments and Suggestions for Authors
I have no additional comments regarding the study and suggest that it can be accepted in its current form.